

# Automated pupillometry to detect command following in neurological patients: a proof-of-concept study

Alexandra Vassilieva[1], Markus Harboe Olsen[2], Costanza Peinkhofer[1,3], Gitte Moos Knudsen[1,4,5] and Daniel Kondziella[1,4,6]

[1] Department of Neurology, Rigshospitalet, University of Copenhagen, Copenhagen, Denmark
[2] Department of Neuroanesthesiology, Rigshospitalet, University of Copenhagen, Copenhagen, Denmark
[3] Medical Faculty, University of Trieste, Trieste, Italy
[4] Faculty of Health Sciences and Medicine, University of Copenhagen, Copenhagen, Denmark
[5] Neurobiology Research Unit, Rigshospitalet, Copenhagen University Hospital and Center for Integrated Molecular Brain Imaging, Copenhagen, Denmark
[6] Department of Neuroscience, Norwegian University of Science and Technology, Trondheim, Norway

## ABSTRACT

**Background.** Levels of consciousness in patients with acute and chronic brain injury are notoriously underestimated. Paradigms based on electroencephalography (EEG) and functional magnetic resonance imaging (fMRI) may detect covert consciousness in clinically unresponsive patients but are subject to logistical challenges and the need for advanced statistical analysis.

**Methods.** To assess the feasibility of automated pupillometry for the detection of command following, we enrolled 20 healthy volunteers and 48 patients with a wide range of neurological disorders, including seven patients in the intensive care unit (ICU), who were asked to engage in mental arithmetic.

**Results.** Fourteen of 20 (70%) healthy volunteers and 17 of 43 (39.5%) neurological patients, including 1 in the ICU, fulfilled prespecified criteria for command following by showing pupillary dilations during $\geq 4$ of five arithmetic tasks. None of the five sedated and unconscious ICU patients passed this threshold.

**Conclusions.** Automated pupillometry combined with mental arithmetic appears to be a promising paradigm for the detection of covert consciousness in people with brain injury. We plan to build on this study by focusing on non-communicating ICU patients in whom the level of consciousness is unknown. If some of these patients show reproducible pupillary dilation during mental arithmetic, this would suggest that the present paradigm can reveal covert consciousness in unresponsive patients in whom standard investigations have failed to detect signs of consciousness.

Corresponding author
Daniel Kondziella,
daniel_kondziella@yahoo.com

## INTRODUCTION

It can be difficult to assess if patients with acute brain injury are conscious by means of standard clinical examinations alone because these patients must be sufficiently aroused and

able to mobilize motor function (*Schnakers et al., 2009a*; *Schnakers et al., 2009b*; *Laureys et al., 2010*; *Di et al., 2014*; *Kondziella et al., 2016*). Thus, standard neurological assessment often misclassifies unresponsive patients as being in a vegetative state (VS, aka. unresponsive wakefulness syndrome, UWS) (*Schnakers et al., 2009b*). This has important implications for prognosis and puts these patients at risk of unjustified withdrawal of life-sustaining therapy (*Demertzi et al., 2011*; *Turgeon et al., 2011*; *Ong, Dhand & Diringer, 2016*; *Harvey et al., 2018*).

Our limited knowledge of disorders of consciousness contributes to this dilemma. It is still not widely recognized that up to 15% of patients are entirely unable to interact with their environment because of complete motor paralysis, despite being minimally conscious (minimal conscious state, MCS) or even fully conscious (*Kondziella et al., 2016*). This state of covert consciousness in completely paralyzed patients has been termed cognitive motor dissociation (*Schiff, 2015*). Owen and co-workers were the first to document cognitive motor dissociation in a landmark paper from 2006 (*Owen et al., 2006*). Herein, the authors showed that a young traffic accident victim without any signs of consciousness at the bedside, thereby fulfilling clinical criteria of VS/UWS, was able to follow commands simply by modulating her brain's metabolic activity as measured by functional magnetic resonance imaging (fMRI) (*Owen et al., 2006*). Thus, in the past 15 years consciousness paradigms based on fMRI and electroencephalography (EEG) that circumvent the need for motor function have been developed (*Monti et al., 2010*; *Cruse et al., 2011*; *King et al., 2013*; *Sitt et al., 2014*; *Stender et al., 2014*; *Rohaut et al., 2017*; *Vanhaudenhuyse et al., 2018*). However, although these technologies may detect covert consciousness, fMRI- and EEG-based paradigms are labor-intensive, expensive, logistically challenging and not readily available in the intensive care unit (ICU) (*Weijer et al., 2015*; *Kondziella et al., 2016*). A cheap and fast, easy-to-interpret point-of-care test for consciousness assessment at the bedside is clearly needed.

Portable infrared pupillometry is a new technology that may prove useful for the determination of covert consciousness in the clinically unresponsive patient. The pupillary reflex is a polysynaptic brainstem reflex under cortical modulation, i.e., cognitive processes such as decision making and mental arithmetic produce pupillary dilation (*Kahneman & Beatty, 1966*; *Kahneman, 1973*; *Loewenfeldt, 1999*; *Marquart & De Winter, 2015*; *Steinhauer, Condray & Pless, 2015*; *Quirins et al., 2018*). Hence, pupillary responses following mental arithmetic have been used to establish communication with patients with the locked-in syndrome and to detect command-following in one patient in MCS (*Stoll et al., 2013*). However, these were patients with chronic brain injury many months or years after the injury, and the technical equipment used was complex, involving a fixed bedside camera and a computer screen for the display of visual instructions (*Stoll et al., 2013*). Here, we wished to assess whether a convenient hand-held pupillometer and a simpler paradigm with vocal instructions allow reliable measurements of pupillary dilation during mental arithmetic as a sign of command following, and hence consciousness, in a wide range of neurological patients admitted for in-hospital care.

## METHODS

### Objectives

We aimed to evaluate a paradigm for the assessment of consciousness and command following in patients with neurological disorders admitted to in-patient hospital care. To this end, we assessed pupillary dilation following mental arithmetic in neurological patients in the ICU and neurological ward, as well as in healthy volunteers. Sedated unconscious patients served as negative controls. We hypothesized that most (but not necessarily all) healthy volunteers and conscious neurological patients would be able to cooperate and show pupillary dilation during mental arithmetic, whereas unconscious and sedated patients would not.

### Study population

We collected a convenience sample of 48 neurological patients admitted to the neuro-ICU and neurological wards at Rigshospitalet, Copenhagen University Hospital, including unsedated ICU patients with spontaneous eye opening in MCS minus (i.e., evidence of visual pursuit; $n = 1$, pupillary dilation measured twice at 7 days interval) or better (conscious state; $n = 1$). Five unconscious and deeply sedated/comatose patients in the ICU were recruited for negative control (Richmond Agitation-Sedation Scale score $-4$, i.e., deep sedation without response to voice, but possibly movement to physical stimulation). Levels of consciousness were estimated following standard neurological bedside examination by a board-certified neurologist experienced in neurocritical care and according to established criteria (*Giacino et al., 2018*). Twenty healthy volunteers served as positive controls.

### Automated pupillometry and mental arithmetic paradigm

The integrity of the pupillary light reflex of both eyes was checked using the NPi®-200 Pupillometer (NeurOptics, Laguna Hills, CA, USA). We documented the neurological pupil index (NPi), which is a proprietary pupillometry sum score from 0–5, with $\geq 3$ indicating physiological limits (including a maximal difference between the 2 eyes of <0.7) (*Chen et al., 2011*; *Larson & Behrends, 2015*; *Peinkhofer et al., 2018*), pupillary diameters before and after light exposure, percentage change of pupillary diameters, and pupillary constriction and dilation velocities. Patients with non-physiological values were excluded. Then, we used the PLR®-3000 pupillometer (NeurOptics, Laguna Hills, CA, USA) to track pupillary size of the right eye over time (approximately 3–5 min in total), while asking the participants to engage in mental arithmetic. During the examination, the examiner held the pupillometer in one hand and covered the opposite eye with the other hand to avoid that changes in ambient light intensities would influence pupil size in different subjects. The set-up was identical for healthy volunteers and patients, except that patients sometimes were examined in the supine position (Fig. 1). Each participant was asked to calculate a series of 5 arithmetic problems of moderate difficulty ($21 \times 22$, $33 \times 32$, $55 \times 54$, $43 \times 44$, $81 \times 82$; approx. 30 s each) with rest periods (30 s) in-between. A subgroup of patients was given arithmetic problems of lesser difficulty ($4 \times 46$, $8 \times 32$, $3 \times 67$, $6 \times 37$, $7 \times 43$; approx. 15 s each, with 15 s rest periods). We carefully explained all participants that pupillary dilation is induced solely by the efforts associated with mental arithmetic,

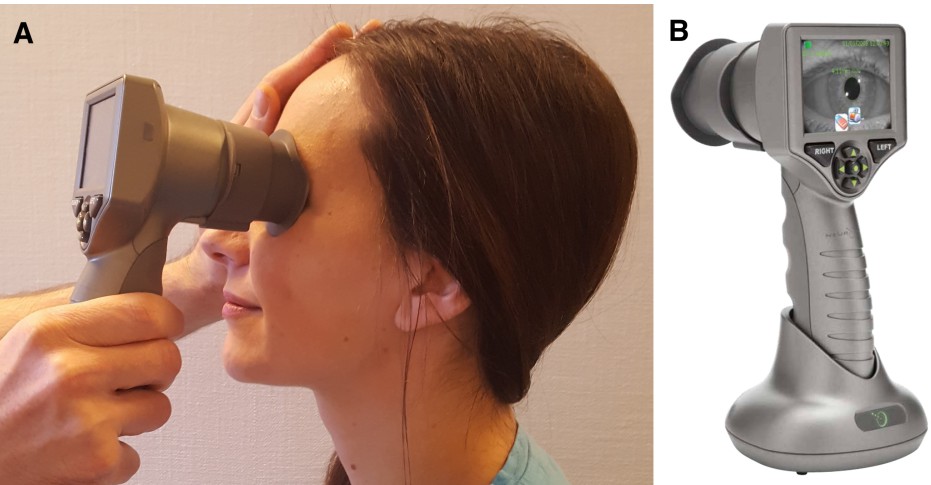

**Figure 1  Pupillary dilation during mental arrithmetic assessed by automated pupillometry.** We used the PLR®-3000 pupillometer (NeurOptics, Laguna Hills, CA, USA), an automated handheld device (A and B), to track pupillary size, while asking patients and healthy volunteers (one shown here; permission obtained) to perform mental arithmetic. The examiner holds the pupillometer in one hand and covers the other eye with the other hand (A). The set-up is identical for healthy volunteers and patients, except that patients may be better examined in the supine position.

and that it was irrelevant for our study if their calculations were correct or not. Hence, participants were instructed not to reveal the results of their mental arithmetic but to pay attention to the task and make an honest effort.

## Outcome measures
Outcome measures included pupillary diameters during periods of mental arithmetic (intervention) and relaxation (rest periods).

## Statistical analysis
Data were analysed using R (*R Core Team, 2017*) by a blinded investigator. Pupillary measurements were visually assessed for quality control in a run chart. Pupillary diameter changes in each of the five mental arithmetic tasks (intervention) were assessed by comparing the period of intervention with the periods of relaxation (rest periods) before and after. Successful pupillary dilation during intervention was defined as a significantly larger median pupillary size during mental arithmetic compared to the immediate rest periods prior and after ($p$-value < 0.01; Wilcoxon signed-rank test, followed by Bonferroni correction). We deemed command following to be successful when a participant showed pupillary dilation in at least four of the five mental arithmetic tasks (80%).

## Ethics
The Ethics Committee of the Capital Region of Denmark approved the study (journal-nr.:H-18045266). Written consent was obtained from all participants or their next-of-kin (unconscious or minimally conscious ICU patients). Data were anonymized and handled according to the European Union's Data Protection Law. The pupillometry device used

**Table 1  This tables shows demographic data, including neurological diagnoses.** "Stroke" includes hemorrhagic and ischemic stroke, "neuromuscular" includes Guillain-Barré syndrome, chronic inflammatory demyelinating polyradiculopathy, Isaacs' syndrome, Pompe's disease and multifocal motor neuropathy, "epilepsy" denotes epilepsy with or without structural cause on magnetic resonance imaging.

| | Healthy volunteers | Neurological patients | Neurological patients[a] | ICU patients, MCS or CS | ICU patients, coma |
|---|---|---|---|---|---|
| N | 20 | 20 | 21 | 2 | 5 |
| Female | 10 (50%) | 9 (45%) | 8 (38%) | 2 (66%) | 2 (40%) |
| Age in years, median (IQR) | 34.5 (29–47) | 60.5 (51–68) | 50 (41–70) | 34 (34–34) | 62 (55–64) |
| Stroke | – | 2 | 2 | 1[b] | 0 |
| SAH | – | 0 | 1 | 0 | 2 |
| TBI | – | 0 | 0 | 0 | 2 |
| Epilepsy | – | 5 | 2 | 0 | 0 |
| Neuromuscular | – | 9 | 10 | 1 | 0 |
| Other[c] | – | 4 | 6 | 0 | 1 |

Notes.

CS, conscious; ICU, intensive care unit; IQR, interquartile range; MCS, minimally conscious state; N, number of subjects; SAH, subarachnoid hemorrhage; TBI, traumatic brain injury.

[a] All mental arithmetic tasks involved $2 \times 2$-ciffered calculations (e.g., $33 \times 32$), except for $1 \times 2$-ciffered calculations (e.g., $8 \times 32$) in neurological patients indicated with (*).

[b] This unsedated ICU patient in MCS with a pontine hemorrhagic stroke was examined twice at 7 days interval but failed to show command following during mental arithmetic in both sessions.

[c] Other diagnoses, not listed above, include relapsing remitting multiple sclerosis, unspecified sensory disturbances, brain abscess, anoxic-ischemic encephalopathy and hemangioblastoma.

in the present study (PLR®-3000 Pupillometer; NeurOptics, Laguna Hills, CA, USA) was on loan from the manufacturer; however, neither the manufacturer, nor the vendor were involved in the design or conduct of the study, data analysis or writing of the manuscript, and the authors did not receive any other monetary or non-monetary benefits.

# RESULTS

We examined 70 participants, two of which were excluded because of a NPi below 3, suggesting abnormal physiological pupillary function. Hence, we enrolled 68 participants: 20 healthy controls, 41 neurological patients on the ward and seven patients in the neuro-ICU. (One patient in the neuro-ICU was measured twice, resulting in 69 assessments in total). Diagnoses reflected a wide spectrum of neurological disorders, including cerebrovascular, neuromuscular, epilepsy, trauma, neuroinfections, and multiple sclerosis. Baseline pupillary function was normal in all participants and did not differ between healthy controls (mean NPi score $4.3 \pm 0.39$) and neurological patients ($4.3 \pm 0.32$; $p = 0.812$). Table 1 shows demographic data.

Pupillary dilation was seen in 65 of 100 (65%) measurements in healthy controls; in 58 of 100 (58%), respectively, 72 of 105 (68.6%, simpler tasks) measurements in neurological patients on the ward; and in seven of 15 (46.67%) measurements in unsedated ICU patients. By contrast, larger pupillary diameters were noted during seven out of 25 (28%) measurements in comatose/sedated ICU patients (negative control group), consistent with chance occurrence.

Fourteen of 20 (70%) healthy volunteers fulfilled the prespecified criteria for successful command following, whereas this was the case for only 16 of 41 (39%) neurological

**Table 2** This table depicts the rate of successful command following by mental arithmetic in healthy volunteers, conscious neurological patients on the ward, minimally or fully conscious patients in the ICU, and comatose/sedated ICU patients. Successful command following was defined by $\geq 4$ significant pupillary dilations during five mental arithmetic tasks.

|  | Healthy volunteers | Neurological patients | Neurological patients[a] | ICU patients, MCS or CS | ICU patients, coma/sedation |
|---|---|---|---|---|---|
| N | 20 | 20 | 21 | 2 | 5 |
| 0 significant | 2 | 3 | 1 | 1[b] | 1 |
| 1 significant | 2 | 1 | 1 | 0 | 2 |
| 2 significant | 0 | 2 | 1 | 1[b] | 1 |
| 3 significant | 2 | 6 | 10 | 0 | 1 |
| 4 significant | 13 | 5 | 4 | 0 | 0 |
| 5 significant | 1 | 3 | 4 | 1 | 0 |
| Successful | 70% ($n = 14/20$) | 40% ($n = 8/20$) | 38% ($n = 8/21$) | 33% ($n = 1/3$) | 0% ($n = 0/5$) |

**Notes.**

CS, conscious; ICU, intensive care unit; MCS, minimally conscious state; N, number of subjects.

[a] All mental arithmetic tasks involved $2 \times 2$-ciffered calculations (e.g., $3 \times 32$), except for $1 \times 2$-ciffered calculations (e.g., $8 \times 32$) in neurological patients indicated with (*).

[b] This unsedated ICU patient in MCS with a pontine hemorrhagic stroke was examined twice at 7 days interval but failed to show command following during mental arithmetic in both sessions.

patients on the ward, one (conscious) of two unsedated patients in the ICU, and 0 of five comatose/sedated patients in the ICU. Healthy controls had higher rates of command following than neurological patients (risk ratio 1.81, 95% CI [1.13–2.99]; $z$-statistic 2.48; $p = 0.013$; excluding sedated ICU patients). Reducing the degree of difficulty of the mental arithmetic task did not change the proportion of neurological patients passing criteria for command following (8/20 patients, 40% vs. 8/21 patients, 38%).

Table 2 provides details. Examples from healthy controls and neurological patients are given in Figs. 2 and 3. Anonymized raw data are available in Data S1–S3.

## DISCUSSION

Cognitive and emotional processes evoke pupillary dilation in both humans and non-human primates, reflecting vigilance, arousal and attention (*Laeng, Sirois & Gredebäck, 2012*; *Schneider et al., 2016*; *Becket Ebitz & Moore, 2017*; *McGarrigle et al., 2017*; *Foroughi, Sibley & Coyne, 2017*). Hence, pupillary diameters may serve as an index of brain activity and mental efforts (or lack hereof) (*Quirins et al., 2018*). Here, we employed mental arithmetic as a paradigm for patients and healthy volunteers to control and maximize pupil dilation to signal command following. We found that a short session of mental arithmetic with simple verbal instructions, without prior training, revealed command following as detected by a handheld automated pupillometry device in 70% of healthy volunteers and 40% of conscious neurological patients.

Seventy % command following in healthy people may seem low, but this figure is very consistent with what has been reported with active EEG- and fMRI-based paradigms. For instance, in one study, 9 of 12 (75%) healthy controls produced EEG data that could be classified significantly above chance (*Cruse et al., 2011*); in another study, 12 of 16 healthy subjects had EEG responses to motor imagery [75.0% (95% CI [47.6–92.7]%)] (*Edlow et al., 2017*). Similarly, 11 of 16 healthy volunteers [68.8% (95% CI [41.3–89.0]%)]

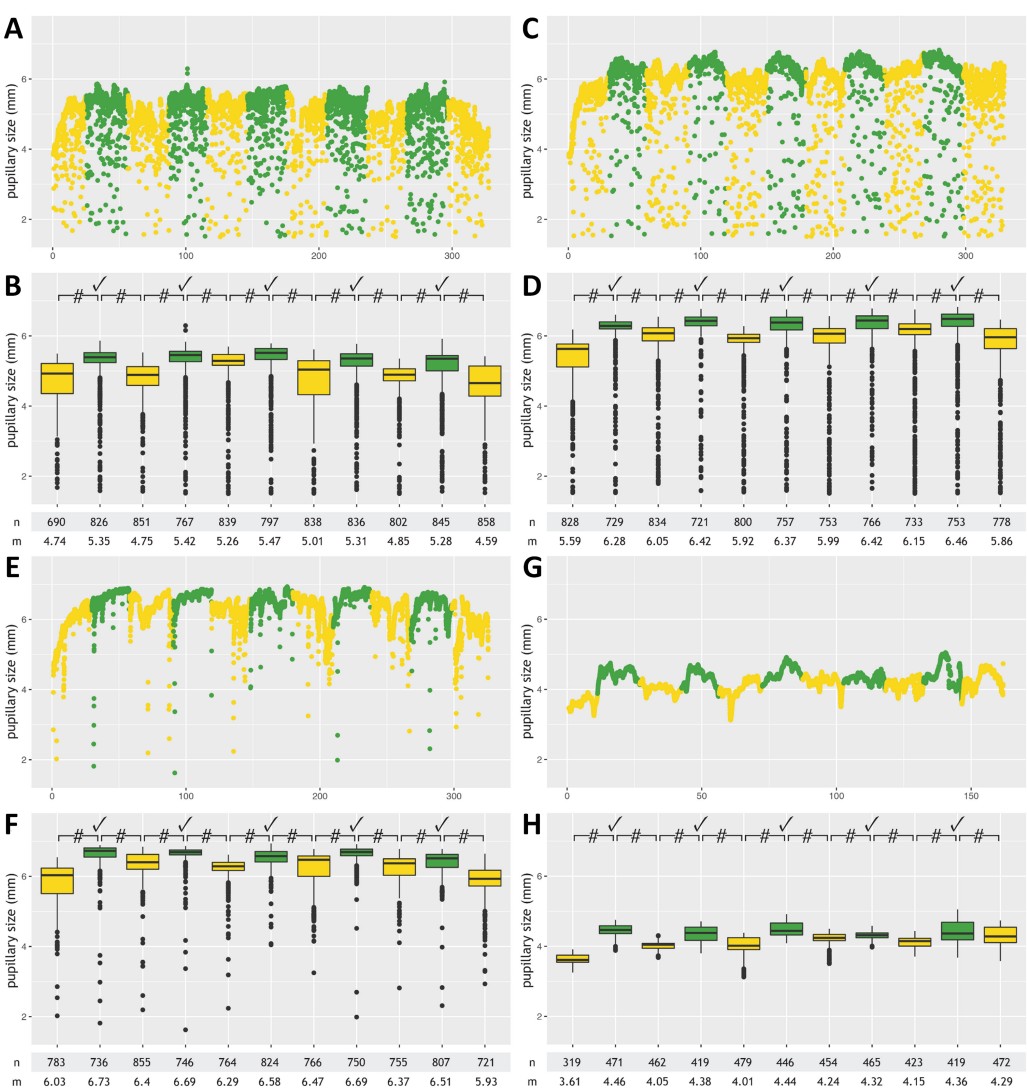

**Figure 2** **Pupillometry data from four patients with successful command following.** This figure shows results from participants with successful command following, detected by automated pupillometry, during a mental arithmetic paradigm: two patients admitted to the neurological ward with diagnoses of multifocal motor neuropathy (A, B) and Guillain-Barré syndrome (C, D), respectively, a healthy participant (E, F), and a conscious 34-year old male with the pharyngeal-cervical-brachial variant of Guillain-Barré syndrome admitted to the ICU (G, H). Data from each subject are presented twice and in 2 different formats (raw measurements A, C, E and G; annotated data B, D, F, H). Minor artifacts due to blinking or eye movements are seen in A, C and E, but not G (probably because of facial and oculomotor nerve palsies). Color code: Periods with mental arithmetic are shown in green, rest periods in yellow. Numbers on the x-axis ("0-100-200-300") denote time in seconds. Pupillary sizes during mental arithmetic were significantly larger (p-value < 0.0001) than during rest periods, consistent with pupillary dilation, in all five tasks for each of the four participants. #, p-value < 0.0001; ✓, pupillary dilation; n, number of measurements; m, median pupillary size; mm, millimeter.

demonstrated responses within supplementary motor areas and premotor cortices when examined by a motor imagery fMRI paradigm (*Edlow et al., 2017*). Again, 7 of 10 (70%) healthy subjects were able to demonstrate covert command following in another motor

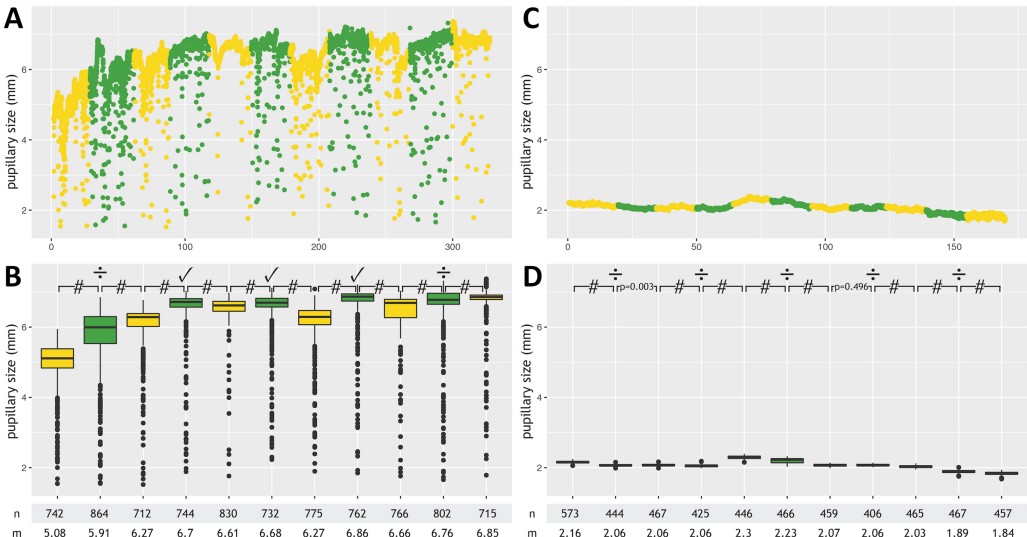

**Figure 3** **Pupillometry data from two patients without command following.** This figure depicts results from a healthy volunteer with unsuccessful command following (A, B) and a 62-year old male in the ICU with subarachnoid hemorrhage and deep sedation (Richmond Agitation-Sedation Scale score of −4) who served as a negative control (C, D). Data from each subject are presented twice and in 2 different formats (raw measurements A and C; annotated data B and D). Minor artifacts due to blinking or eye movements are seen in A, but not B (because of sedation-induced impairment of the blink reflex). In the healthy volunteer, pupillary dilation was noted in only three out of five mental arithmetic tasks, which did not meet our prespecified criteria for successful command following (≥ 4 pupillary dilations, 80%). Minor random fluctuations in the pupillary diameter are seen in the unconscious sedated ICU patient. #, *p*-value < 0.0001; ✓, pupillary dilation; ÷, absence of pupillary dilation; n, number of measurements; m, median pupillary size; mm, millimeter.

imagery fMRI study (*Bodien, Giacino & Edlow, 2017*). Thus, many healthy people cannot cooperate in active paradigms. Of note, however, mental arithmetic seems to generate the most robust activation in the majority of healthy subjects for both EEG and fMRI (*Harrison et al., 2017*). Obviously, in the present study we used relatively difficult multiplication tasks, and easier ones such as serial 7's (*Steinhauer, Condray & Pless, 2015*) might have resulted in a greater fraction of participants being able to comply with our paradigm. Indeed, a few participants who were unable to comply stated that they gave up because they felt stressed by the calculations (although reducing the level of abstraction from 2 × 2-ciffered to 1 × 2-ciffered calculations did not improve compliance). It is interesting in this regard that a recent report showed that it is possible to probe for consciousness using pupillometry also without interfering with participants' stream of consciousness by questioning them, albeit using a much more complex 'local global' auditory paradigm (*Quirins et al., 2018*).

To assess the feasibility of mental arithmetic and pupillometry in the clinical setting, we pragmatically enrolled a large variety of patients with neurological disorders. Not surprisingly, the rate of command following was substantially lower—around 40%—either because of mild cognitive dysfunction related to the underlying neurological condition, the mental stress associated with being admitted to hospital, the higher median age, a lower level of education or a combination of these factors, although we did not examine this

specifically. It is likely that allowing training sessions until patients feel they can cooperate might have yielded a higher success rate. Although we enrolled only two unsedated ICU patients, one of them successfully participated in our paradigm, which corroborates the feasibility of our approach. Also, as expected, none of the sedated ICU patients met our prespecified criteria of command following despite spontaneous fluctuations in pupillary diameters.

Our paradigm has a few limitations that should be discussed. First, one should be aware that, in principle, pupillary dilations can occur simply as a sign of arousal rather than evidence for mathematical calculations and command following. However, to reduce the likelihood for false positive results we employed five serial arithmetic tasks requiring four correct dilations (80%) as evidence for command following. Second, in darkness the pupils will usually be relatively large, which limits the range of pupillary movement (dilation) that might follow a command. Adding a dim light into the pupillometer would constrict the pupil into a range where dilations might be clearer. Third, participants received mental arithmetic tasks not by standardized recorded instructions but by oral instructions delivered by the experimenter; however, we believe that the convenience of oral instructions at the bedside by far outweighs the very slight bias that may possibly arise due to minor deviations in intonations, etc. Of note, statistical evaluations of pupillary data were assessed by a blinded investigator, which strengthens the robustness of our paradigm. Fourth, it should be remembered that medications such as opioids influence pupillary function (and, obviously, consciousness levels). Finally, we did not use an algorithm to remove noise from artifacts caused by head movements, blinking, and partial lid closures (*Larson & Behrends, 2015*; *Neice et al., 2017*; *Behrends et al., 2018*). However, the purpose of our paradigm was not to study pupillary function per se but to identify evidence for covert consciousness using a very simple bedside technique, and we therefore considered algorithms to remove noise unnecessary (and detrimental to the simplicity of the paradigm). Importantly, artifacts from eye or head movements may be more frequent in patients in the MCS than in patients with the VS/UWS and might deserve assessment as consciousness markers in their own right; this should be evaluated by future research.

## CONCLUSIONS

Here we have shown that a fast and easy paradigm based on automated pupillometry and mental arithmetic is able to detect command following in healthy volunteers and conscious patients with neurological disorders admitted to in-hospital care. As a next step, we plan to focus on unsedated non-communicating ICU patients in whom the level of consciousness is unknown. Patients who show reliable pupillary dilation following mental arithmetic are likely to be conscious, whereas absence of pupillary dilation is a poor predictor of lack of consciousness. We suggest that our paradigm can be helpful to identify consciousness in non-communicating patients with acute brain injury for whom traditional bedside examination and laboratory investigations have failed to detect signs of covert consciousness. Compared to EEG and fMRI, pupillometry for the detection of command following, and ultimately covert consciousness, would offer several advantages

in the clinical setting, including quick and convenient assessment at the bedside, simple analysis and low costs.

### Funding
The authors received no funding for this work.

### Competing Interests
The pupillometry device used in the present study (NPi®-200 Pupillometer (NeurOptics, Laguna Hills, CA, USA) was on loan from the manufacturer. Neither the manufacturer, nor the vendor, were involved in the design or conduct of the study, data analysis or writing of the manuscript, and the authors did not receive any other monetary or non-monetary benefits. The authors have no other competing interests.

### Author Contributions
- Alexandra Vassilieva performed the experiments, analyzed the data, approved the final draft.
- Markus Harboe Olsen performed the experiments, analyzed the data, prepared figures and/or tables, authored or reviewed drafts of the paper, approved the final draft.
- Costanza Peinkhofer performed the experiments, authored or reviewed drafts of the paper, approved the final draft.
- Gitte Moos Knudsen contributed reagents/materials/analysis tools, authored or reviewed drafts of the paper, approved the final draft.
- Daniel Kondziella conceived and designed the experiments, performed the experiments, analyzed the data, contributed reagents/materials/analysis tools, prepared figures and/or tables, authored or reviewed drafts of the paper, approved the final draft.

### Human Ethics
The following information was supplied relating to ethical approvals (i.e., approving body and any reference numbers):

The Ethics Committee of the Capital Region of Denmark approved the study (journal-nr.:H-18045266). Written consent was obtained from all participants or their next-of-kin (unconscious or minimally conscious ICU patients).

### Data Availability
The raw data are available in the Supplemental Files and include pupillometry data and basic demographic data (but not age or sex, which have been anonymized in order to comply with the European Union's Data Protection Law).

### Supplemental Information
Supplemental information for this article can be found online at http://dx.doi.org/10.7717/peerj.6929#supplemental-information.

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
