# Peer review of "Automated pupillometry to detect command following in neurological patients: a proof-of-concept study"

_PeerJ, doi:10.7717/peerj.6929_

## Round 0.1 · original submission · Major Revisions

Dear Authors,

We have reports from two peer reviewers who have requested that major revisions be made to your manuscript. PeerJ hopes that you are able to complete these revisions in a timely manner.

·

Basic reporting

With regard to the references, the authors have omitted a recent publication by Steinhauer, SR and Pless, L. (Journal of Ophthalmology, "Pharmacological Isolation of Cognitive...etc"), 2015 that reported pupillary dilations in all volunteers given the task of subtracting sevens. Also in this reviewer's opinion the multiplication tasks are too difficult and might explain why only 70% of the pupils dilated in normal subjects. Many young subjects cannot perform these tasks because they use computers to multiply numbers. Chapter 13 in Irene Loewenfeld's book (The Pupil) discusses why tasks that are too difficult will not be processed by the subject (they give up) and then the pupil does not dilate. The Kahneman reference is fine, but a more complete discussion of the subject is given in his book: Attention and Effort, 1973, Prentice-Hall.

In the opening sentence to the third paragraph of the introduction, this reviewer thinks the sentence: "automated pupillometry....test" should be changed to something like: "Portable infrared pupillometry is a new technology that might provide useful information relating to the determination of consciousness in the paralyzed patient.'

Experimental design

In the Methods section the authors do not state if the opposite eye is covered or not. The Figure 1 seems to show that it is covered. In other words, these measurements were taken in darkness. A pupils of a drowsy subject in darkness will reveal periodic constrictions with subsequent dilations that are related to brief arousals. This should not be an issue in the present study unless some of the subjects were tired. Looking at Figure 2 C, the difference in the diameters between the rest period and the task period seems to show that the rest period was associated with periodic constrictions of the pupil, similar to what one might observe in a drowsy subject. Then when this subject was presented with the mathematical task, these constrictions became less frequent, perhaps because the subject became more alert. Overall, the conclusion for this subject would not be any different from another subject who actually performed mathematical calculations so it would not negate the idea that prompting neocortical activity, for what ever reason, would dilate the pupil of a conscious subject. However, the authors should be aware that some of these positive results might not have involved mathematical calculations, but were simply an arousal response.

There is a minor flaw in the experimental design relating to the fact that the author's have not used an algorithm to remove artifacts cased by head movements, blinks, eye closures longer than a blink, and partial lid closures. Consequently there is noise in the recordings from these artifacts. This does not nullify the overall conclusion. Algorithms to remove noise have been published by Warga, Neice, Bokoch, Meredith. See a recent article by Neice (Anesthesia and Analgesia, 2016, "Prediction of Opioid.....etc) that provides some of these references.
Also in this same article are references to the work of Wilhelm B that discusses the phenomenon of Pupillary Unrest in Darkness as a measure of sleepiness.

Validity of the findings

The findings are valid, but the experimental design needs to be improved for this technique to become widely used.

1. This reviewer is concerned that patients with NPi below 3 were excluded. How many subjects were excluded? If this is a substantial number, then this test would be limited to a small subset of patients.

2. Steinhauer suggests that the dilation of the pupil is also accompanied by a reduction of the light reflex amplitude. The authors might also look at the light reflex amplitude along with pupil diameter as a measure of "command following". Also this same reference suggests that the dilation is primarily brought about by inhibition of the pupilloconstrictor nucleus. If this is true, then subjects who are receiving opioids for sedation might be fully conscious, yet their pupils would not dilate during "command following" because these drugs block inhibitory influences on the PC nucleus (see references by M. Larson).
If it is true that these measurements were taken in darkness, then the pupils will usually be relatively large and this limits the range of pupillary movement (dilation) that might follow a command. One idea would be to add a dim visible light into the pupillometer. This would constrict the pupil into a range where dilations might be more readily apparent. Also if Steinhauer is correct, then the act of "command following" would block the effect of this dim light on pupil size, thereby magnifying the pupillary dilation that would be observed.

Additional comments

This reviewer likes the idea and the presentation of the research. I am concerned that only 70% of normal subjects showed pupillary dilation and that many patients were excluded from the study. There are missing references and the discussion might also include some the above noted comments. Overall with refinement, the technique might be a useful method to determine the conscious state in the paralyzed patient. The authors might also observe that this is a recurring issue in patients who are paralyzed with muscle relaxants and are being artificially ventilated in the intensive care units. So it is a timely subject.

·

Basic reporting

Using a clinical pupillometer probing pupillary dilation in response to cognitive effort, the authors explored the potential for detecting conscious voluntary task engagement in clinically unresponsive patients. They provide interesting preliminary results in controls and patients.

More specifically they show that pupillary dilation in response to task engagement can be observed in:
• 70% of conscious healthy volunteers (similar to previous work)
• between 33 and 40% in clinically conscious or minimally conscious patients
• in 0% of clinically unconscious patients with a very low probability of covert consciousness (e.g., deeply comatose and/or sedated patients)

This approach, although not novel, is interesting because of its technical simplicity opening promising clinical applications.

The manuscript is clear, professional English is used throughout. Background/context and literature references can be slightly improved (see General comments for the authors). The structure of the article, figures and tables match the required standard.
The results of the study are relevant to the hypothesis however, the manuscript requires additional details in the methods section (see General comments for the authors).

Raw data is provided but needs further text and metadata descriptions (see General comments for the authors).

Experimental design

The primary research is within the Aims and Scope of the journal. The research question is well defined, relevant and meaningful. It is stated how this research fills an identified knowledge gap. Investigation was performed with a clinical device with appropriate technical and ethical standards.
The methods description should be improved (see General comments for the authors).

Validity of the findings

The data is robust, well described, statistically sound and controlled. Conclusions are well stated, linked to original research (although this should be developed, see General comments) question and limited to supporting results.

Additional comments

Abstract:
- L45: To avoid misunderstanding by non-expert readers I would specify « clinically unresponsive » in the sentence: “Paradigms based on electroencephalography (EEG) and functional magnetic resonance imaging (fMRI) may detect covert consciousness in unresponsive patients …”
- L57: the term « conscious awareness » is very misleading (synonyms) and should be avoided throughout the MS.

Introduction:
- L70: “Many patients” should be replaced by “some” or simply 15% as in the cited reference in the sentence: “It is still not widely recognized that many patients are entirely unable to interact with their environment because of complete motor paralysis, … »

Methods:
- L106: Study population: more details should be provided for MCS patients. Especially whether these patients were classified MCS only on the visual pursuit item (« MCS-minus ») or on command following item (MCS-“plus”). This could shed more light on the interpretation of the results because in MCS-“minus” this could likely correspond to CMD detection and in MCS-“plus” this result is expected.
- L109: please provide details on criteria for “deeply sedated/comatose patients » (e.g. based on Richmond Agitation-Sedation Scale score for sedation etc..)
- L 128: please discuss whether the instruction to not report the result of the calculation could have affected the sensitivity of the approach. Indeed, pupillary dilation might have been facilitated by the stress related with the possibility of reporting an incorrect result and/or by a greater task engagement, especially in controls.
- Please provide more details on the experimental procedure:
- instruction delivery (standardized recorded instructions or oral instructions delivered by the experimenter?). If there was no standardization please discuss potentials biases due to the absence of blinding in the discussion.
- choice of eye, measurement procedure (experimenter holding the device manually or kind of helmet?)
- How blinks where managed (recording and data processing)?

Result:
- The presentation of the results at the patient level before the task level is a little bit misleading. I wonder if it wouldn’t be easier to read it the other way around.
- Figure 1: would be more useful displaying the patient setting (even with a fake patient).
- Figure 2 & 3 please provide a colored legend. Please also make sure these colors are color blindness compatible. A temporal cue (duration of a bloc or of the entire experiment) would be helpful in these figures.

Discussion:
- The authors should put their results in perspective with the recent work by Quirins et al. Scientific Reports. 2018. Also, some additional previous work on this topic could be cited (see for instance Quirins introduction and discussion).
- Could the authors elaborate about spontaneous ocular movements? Could MCS patients have more spontaneous movements leading to a greater pupil size variability? Could differences in term of visual behavior have interfere with the probed cognitive effect? (e.g. fixation of the experimenter and pupil dilation?).

Raw data:
- The authors provide the raw data, however the supplemental files need more explanation and descriptive metadata identifiers to be useful to future readers.
- Pupillometry data seems to contain 69 individual subjects (but there are 20 controls + 48 patients).
- It also not clear if task/rest periods and multiple measurements info is provided etc…
- In addition, used acronyms should be explained (e.g., « Tid.og.dato » in in peerj-34437-patients_raw) and consistent with the text (e.g., in peerj-34437-patients_raw: “RAS” instead of “RASS”). Finally please ensure there is no irrelevant personal patient data (e.g., date information in « Tid.og.dato » in in peerj-34437-patients_raw).

---

## Round 0.2 · Minor Revisions

Dear Authors

There are still some remaining minor revisions needed, as detailed in the re-reviews.

Thank you

·

Basic reporting

No further comment

Experimental design

No further comment

Validity of the findings

No further comments

Additional comments

1. In the opening sentence of the third paragraph of the Introduction, the word "proof" should be "provide"

2. In the fifth paragraph of the Discussion...beginning with "However, as stated....". My suggestion would be to leave that sentence out because the phrase :"it does not seem to have....", is not data driven. The Neuroptics instrument does permit intensities (1 to 1000 lux) of visible light to be directed into the measured eye, if the protocol asks for it. Perhaps then it would detect "command following" with more consistency, when measuring the midposition pupil, instead of a dilated pupil.

3. This reviewer would like to pass on an idea that has never been tested. It has been known for over 50 years that the sympathetic control of pupil size is lost during states of unconsciousness (see Loewenfeld: THE PUPIL - 1999, and articles by Larson in Anesthesiology - 85:748, 1996). Today there is a remarkable topical agent (Brimonidine,an alpha 2 agonist) that blocks the sympathetic innervation of the dilator muscle. Placing these drops topically into one eye should produce a pharmacologic Horner's syndrome if the immobile paralyzed patient is conscious, but no anisocoria would develop if the patient is unconscious. Just an idea.

·

Basic reporting

The authors have considered all my previous comments in this revised version and I only have very minor suggestions now.

Experimental design

The methods description has been improved. I have no further comments.

Validity of the findings

No further comments.

Additional comments

I think the manuscript has been substantially improved.
The authors have considered all my comments and I have no further major concern.

Minor comments: I think that the word “respectively” in the figure 2 legend is in the wrong place (shouldn’t it be after “(C and D)” ?). In addition, it would be useful to explicitly mention that each subject has data presented twice in 2 different formats. I am also wondering whether the figure 2 and 3 wouldn’t be easier to understand if the 2 plots from a given subject were merged and labeled as one (maybe with boxes to link them, e.g. A and B as A, C and D as B and so on…), but that’s only a suggestion.

The authors should be sure they upload the updated figure legends during the final submission (it is the old one in the attached pdf proof).

Raw data description have been improved but I think the Dataset S3 is still a bit cryptic and would benefit from more explanations. Also please verify data in Dataset S3 (text in cell A1 is truncated, “1” is missing in cell A2 and line 15 only contains “NA”s [is that normal?]).

Sincerely,
Benjamin Rohaut

---

## Round 0.3 · Minor Revisions

Dear Authors,

Please do the minor revisions and resubmit the revised manuscript back.

·

Basic reporting

No further comments

Experimental design

No further comments

Validity of the findings

No further comments

Additional comments

In the Discussion section, lines 212 and following. "Second, physiological pupillary unrest...etc". This sentence is wrong. In darkness, physiological pupillary unrest is essentially absent, unless the subject is very tired, in which case there will be oscillations that appear after several minutes as the subject drifts into a more somnolent state. So the Wilhelm reference states the opposite of what the authors suggest. I would advise that the authors avoid any mention of pupillary unrest.
As a side note however there is a manuscript in the German literature that suggests that a cognitive task increases pupillary unrest when measured in ambient light: "The cognitive pupillary oscillaition hypothesis in patients with neurotic disorders or organic brain syndrome; Reliability and validity of new psychophysiological method" in Wiener Klin Wochenschrift 108:69-74, 1996. In other words, as an alternative to changes in pupil size, it might be possible to observe an increase in pupillary unrest during cognitive tasks in patients who are conscious but paralyzed.

·

Basic reporting

The authors have considered my last comments in this final version.

Experimental design

No further comments.

Validity of the findings

No further comments.

Additional comments

The authors have considered my last comments in this final version.

Of note: double check the ref (there are duplicated author names for Sitt JD., King JR., et al. Brain 2014).

Regards,
Benjamin Rohaut

---

## Round 0.4 · accepted · Accept

Congratulations. Your manuscript has been accepted and will be processed further.

# ·

Basic reporting

No further comments

Experimental design

No further comments

Validity of the findings

No further comments

Additional comments

No further comments

·

Basic reporting

No further comments.

Experimental design

No further comments.

Validity of the findings

No further comments.

Additional comments

No further comments.